# DNA methylation-based epigenetic signatures predict somatic genomic alterations in gliomas

Jie Yang [1,2,3], Qianghu Wang [4], Ze-Yan Zhang [1,2], Lihong Long[5], Ravesanker Ezhilarasan[1,2], Jerome M. Karp[1,2], Aristotelis Tsirigos [6,7], Matija Snuderl[6], Benedikt Wiestler[8], Wolfgang Wick [9], Yinsen Miao[10], Jason T. Huse[11,12] & Erik P. Sulman [1,2 ✉]

Molecular classification has improved diagnosis and treatment for patients with malignant gliomas. However, classification has relied on individual assays that are both costly and slow, leading to frequent delays in treatment. Here, we propose the use of DNA methylation, as an emerging clinical diagnostic platform, to classify gliomas based on major genomic alterations and provide insight into subtype characteristics. We show that using machine learning models, DNA methylation signatures can accurately predict somatic alterations and show improvement over existing classifiers. The established Unified Diagnostic Pipeline (UniD) we develop is rapid and cost-effective for genomic alterations and gene expression subtypes diagnostic at early clinical phase and improves over individual assays currently in clinical use. The significant relationship between genetic alteration and epigenetic signature indicates broad applicability of our approach to other malignancies.

[1] Department of Radiation Oncology, NYU Grossman School of Medicine, New York, NY, USA. [2] Brain and Spine Tumor Center, Laura and Isaac Perlmutter Cancer Center, NYU Langone Health, New York, NY, USA. [3] Quantitative Science Program, MD Anderson Cancer Center UTHealth Graduate School of Biomedical Sciences, Houston, TX, USA. [4] Department of Bioinformatics, School of Biomedical Engineering and Informatics, Jiangsu Collaborative Innovation Center for Cancer Personalized Medicine, Nanjing Medical University, Nanjing, China. [5] Department of Neurosurgery, The University of Texas MD Anderson Cancer Center, Houston, TX, USA. [6] Department of Pathology, NYU Grossman School of Medicine, New York, NY, USA. [7] Applied Bioinformatics Laboratory, NYU Grossman School of Medicine, New York, NY, USA. [8] Department of Neuroradiology, Technical University of Munich, Munich, Germany. [9] German Cancer Research Center (DKFZ) and Department of Neurology and NCT Neurooncology Program, University of Heidelberg, Heidelberg, Germany. [10] Department of Statistics, Rice University, Houston, TX, USA. [11] Department of Pathology, The University of Texas MD Anderson Cancer Center, Houston, TX, USA. [12] Department of Translational Molecular Pathology, The University of Texas MD Anderson Cancer Center, Houston, TX, USA. ✉email: erik.sulman@nyulangone.org

Epigenetics play a crucial role in cancer[1] and show extensive reprogramming through DNA methylation, histone variation, and non-coding RNA. DNA methylation is a stable feature and reflects both inter- and intra-tumor heterogeneity which has been used to classify different types of tumors[2–5]. For example, the recently published DNA methylation-based histopathological classification of central nervous system (CNS) tumors[6] (Unsupervised CNS Classification) has challenged conventional histologic classification and tumor grading. This unsupervised CNS classification used the unsupervised learning approach to identify CNS tumor classes with distinct DNA methylation profiles. The established random forest-based classifier can classify CNS tumor into one of those histopathological classes based on the tumor DNA methylation profile.

Infiltrating gliomas, including WHO grade II-IV gliomas, are the most common and lethal primary brain tumors[7]. These devastating tumors have been subjected to comprehensive, molecular profiling, particularly by The Cancer Genome Atlas (TCGA). Previous studies have identified some key molecular features in gliomas which play critical roles in glioma initiation, progression, diagnosis, and treatment. For example, isocitrate dehydrogenase (IDH) mutation is positively associated with younger age and longer survival time. Chromosome 1p/19q co-deletion (chr1p19q codel) is prognostic for improved survival and predictive of response to chemotherapy[7,8]. IDH mutation and chr1p19q codel are part of the current WHO diagnosis criteria for gliomas. Telomerase reverse transcriptase promoter (TERTp) mutations and alpha thalassemia/mental retardation syndrome X-linked (ATRX) mutations are mutually exclusive alterations in gliomas and both are functionally correlated with telomere length maintenance[9]. Telomerase inhibitory therapies, heterochromatin silence-mechanism targeted therapies, and G4-destabilizing therapies are promising therapeutic targets for gliomas with ATRX or TERTp mutation[10,11]. The $O^6$-methylguanine DNA methyltransferase (MGMT) promoter methylation status[12] is a prognostic and alkylating chemotherapy-predictive biomarker. It can be predicted using an established methylation array-based algorithm MGMT-STP27[13]. At the transcriptional level, glioblastoma (GBM) has been classified into three subtypes based on characteristic gene expression signatures called classical (CL), proneural (PN), and mesenchymal (MES)[14]. CL GBMs are characterized by epidermal growth factor receptor (EGFR) amplification while MES GBMs are enriched for neurofibromin 1 (NF1) deletion and mutations[14].

All somatic alterations described above are essential for diagnosis, treatment decision-making, and patient prognosis. However, individual assays are usually required to obtain each of these somatic alterations. Detection of somatic mutations, such as IDH, ATRX, and TERTp status, often rely on next generation sequencing (NGS). Fluorescence in situ hybridization (FISH) or loss of heterozygosity (LOH) analysis are usually applied to obtain chr1p19q status. Methylation-specific PCR (MS-PCR) or pyrosequencing assays are usually utilized[15] to obtain MGMT promoter status. No clinical assay currently exists to classify tumors by gene expression. This is because the most commonly available analyte following glioma resection is formalin-fixed and paraffin-embedded (FFPE) tissue. FFPE-derived RNA is highly degraded and chemically modified, therefore, its transcriptional sequencing quality is low and can be biased by artifacts and technical variance[16]. In summary, the cost, time, and tissue requirements for these individual assays frequently result in delayed or incomplete molecular diagnosis, leading to suboptimal treatment and ineligibility for clinical trials. Indeed, there is an urgent need to develop a rapid, cost-effective assay that requires minimal amounts of FFPE tissue for patients with infiltrating gliomas.

In our study, we develop a DNA-methylation-based classifier for gliomas, which we validate using an independent cohort. The developed models can be easily applied to all infiltrating gliomas, including both low-grade gliomas and GBM. Exploration of DNA methylation-based misclassified cases provides valuable ideas for future research directions and demonstrates potential superiority of the methylation approach over existing individual assays.

## Results

We aimed to develop a DNA methylation-based classifier which accurately determines IDH, TERTp, and ATRX mutation status, chr1p19q codel status, and gene expression subtype of infiltrating gliomas. The above molecular features can be separated into two categories in terms of their status: binary class, including IDH, TERTp, and ATRX mutation or wild type, and chr1p19q codel or intact; and gene expression subtypes, including CL, PN, or MES. Separate classifiers were developed for prediction of each of the binary classes (IDH, TERTp, etc.) and for prediction of gene expression subtype, using a rigorous machine learning approach. The binary genomic alteration classifiers were trained and validated on a large cohort of both low-grade and high-grade glioma samples from TCGA, while the gene expression subtype classifier was trained on TCGA glioblastoma samples only, since these subtypes were originally described using high-grade glioma datasets. The performance of all classifiers was validated with an independent cohort (NOA-04) from a multicenter phase III randomized trial conducted by the German Neurooncology Working Group (NOA) of the German Cancer Society, which includes both low-grade and high-grade gliomas[17].

**Predictive Models**. For binary genetic alterations, all predictive models achieved high prediction accuracy as shown in Fig. 1A. In the test set, models achieved a prediction accuracy of 100%, 98.31%, 90.48%, and 99.21% for IDH, TERTp, and ATRX mutation, and chr1p19q codel status, respectively, with AUC of 1.0, 1.0, 0.9952 and 0.9974 respectively. For gene expression subtype prediction, GBM samples with HM450K and HM27K data were processed as described in Methods. The final random forest model was refitted with training ($n = 212$) and development ($n = 72$) sets and achieved a prediction accuracy of 72.2% (52/72) in the test set.

**Predictive signature analysis**. For each binary genetic alteration, we filtered out a subset of probes that are statistically different between the binary classes, and then performed the clustering analysis with all samples available (Fig. 1B). Samples were clustered into two subgroups and showed high consistency with the known genomic alteration. By comparing the signature probes of IDH, TERTp, ATRX, and chr1p19q codel with the Glioblastoma-CpG island Methylator Phenotype (G-CIMP) signature[18], we found no significant overlap among these five probe signatures (Fig. 1C). The lack of overlap between ATRX and TERTp mutation signatures was consistent with the mutually exclusive nature of ATRX and TERTp in telomere maintenance[19].

Signatures for binary genetic alterations and gene expression subtypes were summarized for their genetic context enrichment. By comparing the number of probes enriched for chromosomes after normalization, we found that probes in the IDH mutation prediction model were enriched in chromosome 22 (13.08%) and chromosome 21 (8.8%) while probes in the ATRX mutation prediction model were enriched in chromosome 9 (7.2%) and chromosome 14 (7.2%). Interestingly, probes in the TERTp mutation, chromosome 1p/19q co-deletion, and gene expression subtype prediction models were all enriched in chromosome 18 (TERTp: 8.2%; co-deletion: 16.3%; gene expression subtype:

A

| Genomic alterations | #probes in model | Gold Standard | #total samples | Prediction Accuracy (#sample) | | |
|---|---|---|---|---|---|---|
| | | | | Training set | Development set | Test set |
| *IDH* | 100 | DNA-seq | 637 | 100.00%(383/383) | 100.00%(127/127) | 100%(127/127) |
| *TERT*p | 1000 | PCR-seq | 298 | 100.00%(179/179) | 96.67%(58/60) | 98.31%(58/59) |
| *ATRX* | 500 | DNA-seq | 637 | 97.12%(372/383) | 85.16%(109/128) | 90.48%(114/126) |
| Chr 1p/19q | 100 | SNP6 | 641 | 99.74%(384/385) | 97.67%(126/129) | 99.21%(126/127) |

B

C

**Fig. 1 Methyl-based predictive models' performance and signatures analysis. A** Summary of DNA methylation-based model performance for each binary genetic alteration and number of probes involved in each predictive signature. **B** Heatmaps of DNA methylation level (β value): samples are in columns and DNA methylation probes are in rows. The two top sidebars show the sample source and genetic alteration status. From left to right, each image shows the most significant probes in signatures of IDH (number of probes = 100), *TERT*p (number of probes = 200), *ATRX* (number of probes = 90), and chr1p19q codel (number of probes = 70). **C** In this Venn diagram, the predictive model probes of each binary genetic alteration (number of probes: IDH = 100, *TERT*p = 1000, *ATRX* = 500, chr1p19q codel = 100) and the G-CIMP probes (number of probes = 818) identified in the published paper[33] were compared with each other. Different colors represent different probe sets. The overlapping blocks between any probe sets indicate the overlapping probes. The number within the diagram indicates the number of probes within a specific block.

12.6%) (Supplementary Data 1). Summarizing the dispersion of probes in terms of CpG island relationship, we found that most of the probes were enriched on CpG islands. Among the four predictive models, the IDH predictive signature showed the highest percentage of CpG islands (76%). For the other three predictive signatures, about 32% of probes were located on CpG islands (Supplementary Table 1).

Among the 100 probes in the IDH predictive signature, 45% (45/100) were located on the promoter region (including TSS200, TS1500, and 1st exon) (Supplementary Table 2). In total, 65 genes were mapped by IDH signature probes. Among all genes, the CASP8 and FADD like apoptosis regulator (*CFLAR*) gene had four probes mapped and nuclear receptor subfamily 4, group A, member 1 (*NR4A1*) had three probes mapped. Applying those 65 genes to the DAVID[19] (version 6.7) for functional annotation, the top GOs were regulation of apoptosis (*p* value = 0.0052), regulation of programmed cell death (*p* value = 0.0056), and regulation of cell death (*p* value = 0.0057) (Supplementary Data 2). Genes related to those GOs were *CFLAR*, *TNF* receptor superfamily member6 (*FAS*), potassium voltage-gated channel interacting protein 3 (*KCNIP3*), death effector domain containing 2 (*DEDD2*), lectin, galactoside-binding, soluble, 1 (*LGALS1*),

*NR4A1*, proline dehydrogenase 1 (*PRODH*), retinoic acid receptor, gamma (*RARG*), and erb-b2 receptor tyrosine kinase 2 (*ERBB2*). Those GOs were not significant after *p* value adjustment due to the small gene set.

For the *TERT*p predictive, most of the probes were located at the body (29.1%) (Supplementary Table 2). Probes mapped to 612 genes in total. The most frequently mapped gene was isthmin 1 (*ISM1*) with ten probes. The second most frequently mapped gene was atlastin GTPase 3 (*ATL3*) with seven probes. For gene functional annotation, the top significant GOs were all related to regulation of transcription (Supplementary Data 2).

For the *ATRX* predictive signature, most of the probes were located at the body (13.9%) (Supplementary Table 2). In total, probes mapped to 333 genes. Gene F-box protein 6 (*FBXO6*) has five probes mapped and cathepsin F (*CTSF*) gene was mapped four times. Using the DAVID gene functional annotation, genes were significantly enriched in development related GOs, including embryonic development and neuron development (Supplementary Data 2). We further compared the 70 probes overlapped between *ATRX* predictive signature and *TERT*p predictive signature; 52 genes were mapped. Most of the genes were enriched in transcription regulation. The most frequently mapped

gene was F-box protein 6 (*FBXO6*) and phosphodiesterase 7B (*PDE7B*). The top GO enriched for those overlapped genes was cell-cell signaling (Supplementary Data 2).

For the chromosome 1p/19q co-deletion prediction signature, 44% of probes mapped to the promoter region, including TSS200, TSS1500, and 1st Exon (Supplementary Table 2). Four probes mapped to gene *ATL3* and fibroblast growth factor receptor 2 (*FGFR2*). The top two GOs were regulation of cellular protein metabolic process (*p* value = 0.001) and BMP signaling pathway (*p* value = 0.0098). However, no GOs were significant after Benjamini *p* value adjustment (Supplementary Data 2).

For the gene expression subtype prediction signature, most of the probes were mapped to the 1st Exon (39.95%) and 5′-UTR regions (25.8%) (Supplementary Table 3). Four probes mapped to the gene *SOCS2* and three probes mapped to *ERBB2* and *RBP1*. The top two significant GOs were correlated with neuron development and differentiation (Supplementary Data 2).

**Prediction results analyses**. For *ATRX* mutation status, five sample subsets (set 1–5) were formed based on the DNA-seq based *ATRX* status, methyl-based *ATRX* status, and single nucleotide variation (SNV) information (Fig. 2A). Twenty-five samples were classified as wildtype by DNA-seq but mutant by the methyl-based model (Supplementary Table 3). Among these 25 samples, 17 samples (set 2) showed at least one mutation call and eight samples (set 3) had no mutation calls according to the SNVs (Fig. 2B). For set 4, samples with *TERT*p mutation status, 3/8 were *TERT*p mutant and *ATRX* wildtype. All samples misclassified as *ATRX* mutant by methyl-based model harbored IDH mutations while all samples misclassified as wildtype by methyl-based model were IDH wildtype (Fig. 2B). Mutation type shift occurred between set 2 (samples are *ATRX* DNA-seq wild type, methyl-based mutant, and with SNV calls) and set 4 (samples are *ATRX* DNA-seq mutant, methyl-based wild type, and with SNV calls): the enriched mutations shifted from "frameshift indels, in frame indels, and splice site" to "intron, missense and nonsense" which may not lead to *ATRX* loss of function. More importantly, no significant differences in *ATRX* gene expression were observed among set2, set3, and set5, which are all methyl-based *ATRX* mutant samples, and no difference between set1 and set4, which are all methyl-based *ATRX* wild type samples. Interestingly, when the methylation results were discordant, even when the sequencing results were in an agreement, a significant difference in expression was observed (Fig. 2C-D). The DNA methylation level of probes located on *ATRX* did not show significant differences among the three subsets (set2, set3, and set4) except for one probe (Supplementary Fig. 1).

For chr1p19q codel status prediction, five samples were misclassified when comparing methyl-based status to SNP6-based status (Fig. 2E). The CNV profile of chr1 and chr19 were derived from the HM450K methylation data using the R package conumee[20] (Fig. 2F). Four out of five samples were misclassified as codel and one sample was misclassified as non-codel by methylation model. We can clearly observe the deletion in the TCGA-CS-5394 and TCGA-FG-7637 which matches with the methyl-based model prediction. The CNV profile pattern of the other three samples is not obvious; therefore, it is difficult to determine their status.

For gene expression subtype prediction, samples in the test set (*n* = 72) were categorized by methyl-based and transc-based gene expression subtypes (Fig. 3A). Discordant samples between the two methods showed significant difference in copy number variation and gene expression level compared to samples with concordant subtypes. We examined enriched alterations of specific subtypes in discordant samples to determine which classification approach showed the highest association with these

characteristic alterations (Wilcoxon rank sum test) (Fig. 3B-C). This further favors the methyl-based classification among discordant cases classified as CL by transcription and MES by methylation or classified as MES by transcription and CL by methylation.

**Model validation**. The prediction accuracy for each binary genomic alteration in the NOA-04 cohort was: for IDH mutation, 89.9% (98/109) by PCR-seq and 99.10% (114/115) by unsupervised clustering analysis of the HM450k DNA methylation profile; for *TERT*p mutation, 82.8% (82/99) by PCR-seq; for *ATRX* mutation, 92.7% (89/96) by immunohistochemistry (IHC); and for chr1p19q status, 88.89% (88/99) by MLPA and 95.65% (110/115) by HM450K-derived CNV profiles (Fig. 3D). In terms of the IDH mutation status, 11 samples were misclassified by methyl-based prediction: 9/11 were predicted as wildtype by PCR-seq but mutant by the methyl-based model. *MGMT* methylation status comparison is shown in Supplementary Table 4.

In TCGA LGG samples, gene expression subtypes predicted by methyl-based and transc-based algorithms showed large differences in the classification results of the PN subtype (Fig. 3E): 422/486 (86.8%) samples were classified as PN by methyl-based subtype while only 228/486 (46.9%) were classified as PN by transc-based subtype. The heatmap in Fig. 3F aligns methyl-based subtype and transc-based subtype with other key features of gliomas, including histology, 1p1q codel status, *MGMT* promoter methylation status, and mutation and CNV status of critical genes. It is clear that the almost all *IDH1/IDH2* mutations and most *TP53* and *ATRX* mutations matched with the methyl-based PN subtype. *EGFR* amplifications are hardly observed in PN subtype. These observations follow the known characteristics of the PN subtype and support the methyl-based classification.

**UniD vs. Unsupervised CNS classification comparison**. The key difference between the UniD and Unsupervised CNS classification is that UniD aims to predict each infiltrating glioma's key molecular features based on the DNA methylation values of selected loci for each molecular feature, while the Unsupervised CNS classification aims to classify each CNS tumor into one histopathological class based on its overall DNA methylation profile. Accordingly, the Unsupervised CNS classification is an unsupervised learning-based model that has been developed to include all CNS tumors, while UniD is a supervised learning-based model which focuses only on gliomas. The objectives of these two classifiers are different, but it is informative to compare the results of these two classifiers.

Gliomas (*n* = 644) were classified into nine groups based on the UniD predicted molecular features status. These groups and their Unsupervised CNS classification-based classes are summarized. in Fig. 4A. Most gliomas fall into five groups (Grp1, 2, 3, 7, and 8). Gliomas in Grp8 show wild type status in both *ATRX* and *TERT*p, suggesting that alternative mechanisms may exist to maintain their telomere length. Discordant samples between the two classification systems are described in Fig. 4B. First row: 40/644 gliomas were classified as CONTR categories (classified-normal) which are normal brain tissue according to Unsupervised CNS Classification, while the remaining cases were classified into "tumor" categories (classified-tumor). By comparing the ABSOLUTE tumor purity between the classified-normal and classified-tumor samples (Fig. 4C), many classified-normal samples show high tumor purity and 48 classified-tumor samples show tumor purity equal to or lower than the median tumor purity of classified-normal samples. Second row: all CONTR, HEMI (methylation class control tissue, hemispheric cortex) in subgroup 1 to 4 are expected to be IDH wildtype while all have been detected with IDH mutation by DNA sequencing. Third

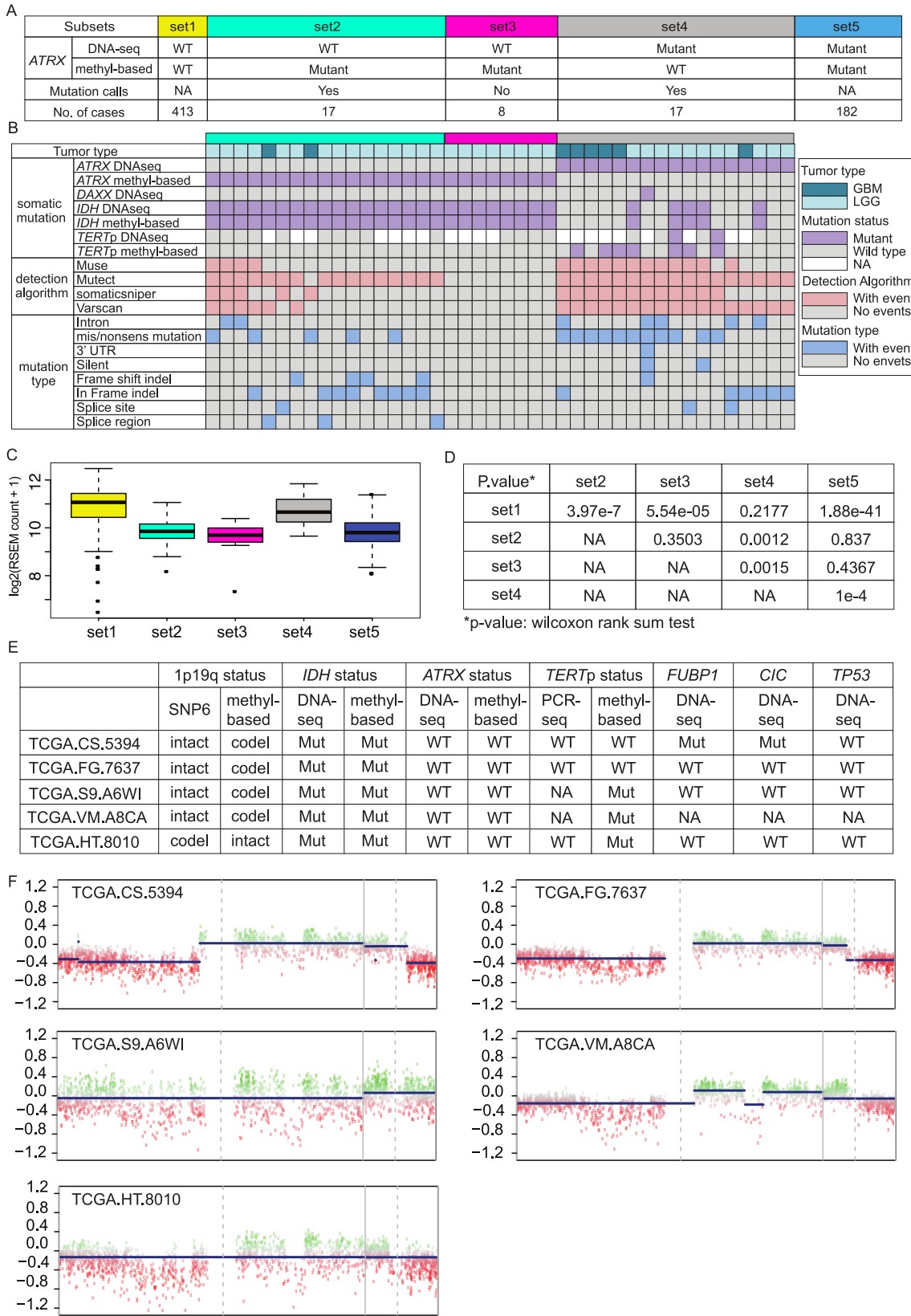

row: Twelve samples in Grp2 (CONTR, HEMI; A IDH; and A IDH, HG) were classified as either normal brain normal tissue or IDH wildtype glioma without chr1p19q codel while their CNV profile from SNP6 showed clear chr1p19q codel (Fig. 4D). Fourth row: SFT, HMPC (methylation class solitary fibrous tumor / hemangiopericytoma) samples are expected to have a euploid genome while

TCGA-19-5951 in Grp7 showed significant chr10 loss and chr19p and chr20 amplification (Fig. 4E). Fifth row: A IDH, HG in Grp8 are expected to be IDH mutant by Unsupervised CNS Classification but, in fact, were wild type by sequencing. Sixth row: Two samples from adult patients (TCGA-06-5858 and TCGA-06-6698) were classified as IHG (infantile hemispheric glioma) by Unsupervised CNS

**Fig. 2 Investigation of misclassified samples for *ATRX* and chr1p19q codel. A** All samples ($n = 637$) used to build methyl-based predictive model for *ATRX* are classified into five subsets based on DNA-seq and methyl-based *ATRX* mutation status and on whether they had mutation calls by reviewing the SNV information. **B** The misclassified 42 samples are shown with their tumor type, mutation status, mutation calling algorithms, and detailed mutation type. **C** Boxplots showed the *ATRX* gene expression level for each subset. Box plot shows center line as median, box limits represent 25th and 75th percentiles, whiskers as 1.5 interquartile ranges above and below box limits or maximum/minimum, whichever is closest to median. By applying the Wilcoxon rank sum test, set 2 and set 3 samples show significantly lower ATRX expression levels than set 1 (set 2 versus set 1: $P$ value $= 3.97 \times 10^{-7}$; set3 versus set 1: $P$ value $= 5.54 \times 10^{-5}$) and set 4 showed significantly higher ATRX expression level compared to set 5 ($P$ value $= 1 \times 10^{-4}$). All $P$ values are two-sided. **D** A two-sample $t$ test was applied to compare *ATRX* gene expression level between every two subsets. The two-sided $P$ value is provided for each comparison in the table. **E** Genetic characterization of the five misclassified samples by comparing methyl-based chr1p19q status to SNP6-based chr1p19q status. **F** CNV profiles of chr1 and chr19 of the five misclassified samples were derived from HM450K data using the R package conumee[25].

Classification which is typically limited to infants. Among the major five groups (Grp 1, 2, 3, 7, 8), the Kaplan-Meier plot and the $P$ values of the log-rank test between any two groups are shown in Fig. 4F, G. No significant differences were observed among Grp1, Grp2, and Grp3. This indicates that all patients with IDH mutant tumors have similar survival regardless of their tumors' *ATRX*, *TERT*p, or chr1p19q status and IDH wildtype glioma (Grp7 and Grp8) showed significantly worse survival compared to IDH mutant gliomas. Moreover, Grp7 demonstrated poorer survival compared with Grp8, whose samples harbored *TERT*p mutations, indicating the negative prognostic significance of *TERT*p mutation in the absence of IDH or *ATRX* mutation. However, the lack of stratification based on *CDKN2A* deletion status, a recently identified prognostic factor[21], may limit the interpretation of these results. Grp7 (75/176, 42.61%) also included a significantly higher percentage ($P < 0.01$) of tumors with *MGMT* promoter methylation compared to Grp8 (7/39, 17.95%).

## Discussion

Our study demonstrates that the DNA methylation microarray-based classifier UniD accurately predicts somatic genomic alterations in infiltrating gliomas and shows improved enrichment for characteristic genomic alterations. Moreover, it is suitable for FFPE samples and can be easily applied to the currently available EPIC array, which contains >850,000 probes. This rapid, low-cost platform outperforms multiple individual assays.

The methyl-based *ATRX* prediction model identified cases with likely loss of function, even when reported as wildtype by sequencing. The comparison between mutation status of *ATRX*, IDH, and *TERT*p and *ATRX* expression level indicates that the methyl-based prediction model can identify tumors with true *ATRX* loss of function with higher accuracy. It is reasonable to speculate that samples in set3 (DNA-seq wild type, methyl-based mutant, and no SNV calls) may be deactivated by some other mechanisms while *ATRX* mutations at the DNA sequencing level in set4 (DNA-seq mutant, methyl-based wild type, and with SNV calls) do not affect the function of the protein; however, this conclusion may be limited due to the small number of cases in each set. When we compared the overall classification based on mutation status of IDH, *ATRX*, and *TERT*p, we found that the methyl-based annotations provided more precise genetic characterizations than DNA-seq annotations (Supplementary Fig. 2).

Circular binary segmentation (CBS) has been used to derive the CNV profile from HM450K methylation data and was applied with the R package conumee[20]. However, this method does not specifically identify partial deletions in chromosome1p and 19q and requires a manual, subjective threshold which is susceptible to considerable inter- and intra-observer variability (Supplementary Fig. 3). The methyl-based predictive model provides an objective determination of CNV status, less subject to this variability.

Currently, there is no clinical assay of transcription-based gene expression subtype determination and as a result the clinical significance of this component of molecular classification has been largely ignored. For example, PN gliomas are more sensitive to chemotherapy and radiation treatment and *NF1*-silenced gliomas were shown to be more sensitive to radiation than temozolomide treatment[22]. MES gliomas have been associated with poor survival and with advanced patient age[14]. Several studies have demonstrated that PN tumors will shift to MES subtype at recurrence and the master regulators of the MES may serve as therapeutic targets[23,24]. This study demonstrates that UniD provides gene expression subtype determination in a clinical setting and facilitates the use of this classification for therapeutic development.

The Unsupervised CNS Classification[6] has shown great value in standardizing and clarifying brain tumor diagnosis by reducing the inter-observer variance and classifying tumors previously unclassified by histology. However, this classification focused on comprehensive CNS tumor classification based on WHO entities and not focused on genomic alterations of specific tumor types, particularly gliomas. Our comparison of Unsupervised CNS Classification and genomic alterations shows that it may not be able to properly identify the specific genomic alteration subclasses studied in this report.

Our classifier demonstrates that DNA methylation signatures accurately predict somatic genomic alterations in human gliomas, emphasizing the extensive and significant relationship between cancer epigenetic signatures and somatic genomic alterations. Given that all predictors are based on a single experimental platform, the Infinium methylation BeadChip arrays, the classifier lends itself to the clinical diagnostic setting. The array's cost, processing time, and tissue requirements are significantly less than individual sequencing, IHC, and copy number assays (for example, FISH) which are currently used clinically. Moreover, the Infinium array is suitable for FFPE samples and can be used for clinical diagnostic tissues.

Besides DNA methylation levels, many array-based bioinformatics tools have been developed, such as MGMT-STP27[13], InfiniumPurify[25] (which estimates tumor purity), and ChAMP-CNV[26], that allow for further unification of glioma biomarkers into a single assay. Lastly, the successful development of this DNA methylation-based, infiltrating glioma-specific classifier highlights that methylation-based tumor classification systems can be easily developed for other tumor types, not only for genomic alteration-based classification, but further grading and prognosis, such as for breast[3] or lung cancer[27].

## Methods

This study was performed in accordance with the guidelines and policies and with approval of the NYU Langone Institutional Review Board (IRB). Based on our institution's policy, our research which uses only non-identifiable data is not considered as research involving human subjects, therefore an IRB approval is not required.

**Data processing**. A total of 129 samples from the TCGA-GBM dataset and 516 samples from the TCGA-LGG dataset were used to train the classifier. Clinical characteristics of the patients from whom the samples were derived are listed in Supplementary Table 5.

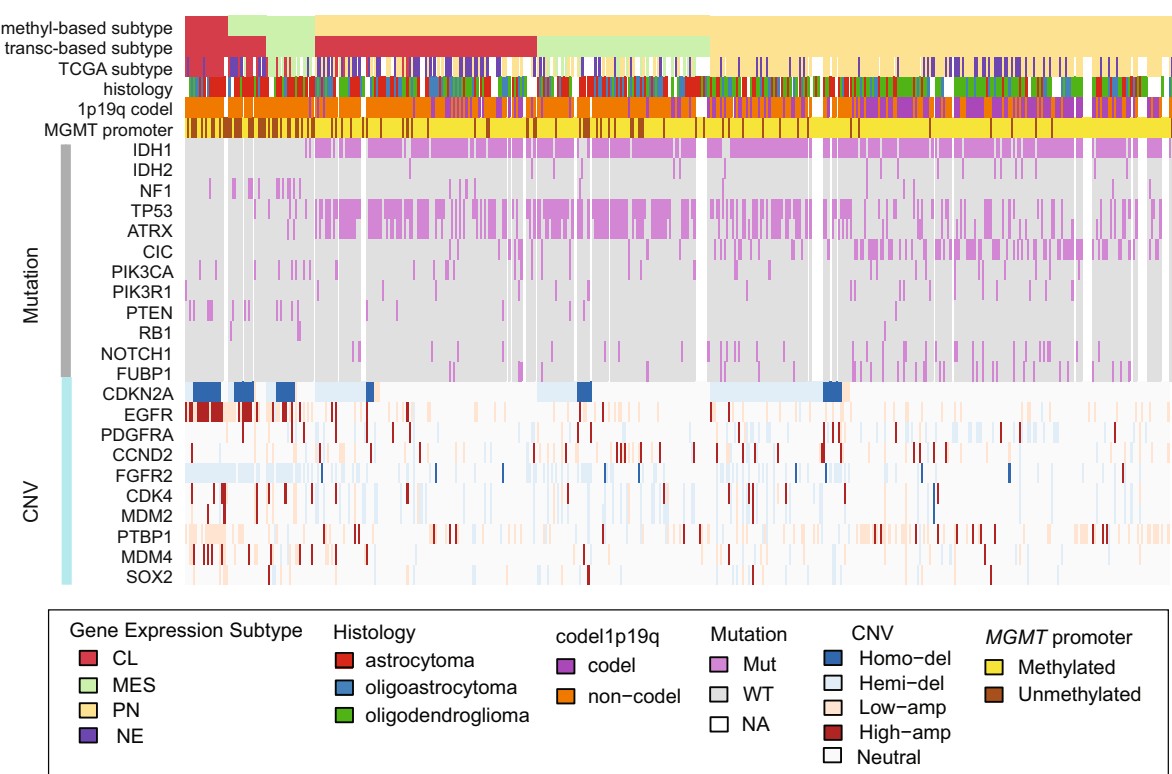

**A**

Comparison of samples
L: transc-based CL & methyl-based CL
R: transc-based CL & methyl-based MES

| test set (n=72) | | Methyl-based subtype | | |
|---|---|---|---|---|
| | | CL | MES | PN |
| Transc-based subtype | CL | 22 | 5 | 1 |
| | MES | 9 | 13 | 0 |
| | PN | 4 | 1 | 17 |

Comparison of samples
L: transc-based MES & methyl-based CL
R: transc-based MES & methyl-based MES

**B** — P-value=0·08 / P-value=0·0056 / P-value=0·015 / P-value=0·029

**C** — P-value=0·04 / P-value=0·00019 / P-value=0·02 / P-value=0·9

**D**

| Genomic alterations | #samples | gold standard | Prediction Acc |
|---|---|---|---|
| IDH | 108 | PCR-seq | 89·90% |
| | 115 | 450k-based clustering | 99·10% |
| *TERT*p | 99 | PCR-seq | 82·80% |
| *ATRX* | 96 | IHC | 92·70% |
| Chr 1p19q | 99 | MLPA | 88·89% |
| | 115 | 450k-derived CNV | 95·65% |

**E**

| validation set TCGA-LGG | | Methyl-based subtype | | |
|---|---|---|---|---|
| | | CL | MES | PN |
| Transc-based subtype | CL | 21 | 19 | 109 |
| | MES | 0 | 24 | 85 |
| | PN | 0 | 0 | 228 |

**F**

Gene Expression Subtype: CL, MES, PN, NE
Histology: astrocytoma, oligoastrocytoma, oligodendroglioma
codel1p19q: codel, non-codel
Mutation: Mut, WT, NA
CNV: Homo−del, Hemi−del, Low−amp, High−amp, Neutral
*MGMT* promoter: Methylated, Unmethylated

**Probe selection**. For genetic alterations within the binary classes, including IDH, *TERT*p, and *ATRX* mutation, and chr1p19 codel, raw data (IDAT files) of infiltrating gliomas profiled on Infinium HumanMethylation 450K BeadChip arrays (HM450K, Illumina) were subjected to sample level and probe level quality control (Fig. 5A). After quality control and probe filtering, one LGG sample did not pass the sample quality control and was excluded from the data set. After probe filtering, the final data set included 644 gliomas samples with 380010 probes. Among the 644 glioma samples, 637 of them had IDH and *ATRX* mutation status available, 298 samples had *TERT*p mutation status annotated, and 641 of them had chromosome 1p/19q co-deletion status annotated.

For all binary genomic alterations, including IDH, *ATRX*, and *TERT*p mutations, and chr1p19q codel, the following filters were applied to the probes available in the Infinium HumanMethylation BeadChip 450 K (HM450K) array: (1) remove probes not mapped to the autosomal chromosomes; (2) remove probes

**Fig. 3 Investigation of misclassified samples for gene expression subtype predictions analysis in test set and gene expression subtype model validation with NOA-04 and TCGA LGG. A** Confusion matrix of the test set ($n = 72$) based on gene expression subtype of transc-based prediction and methyl-based prediction. (L: left; R: right). **B** "Transc-based CL and methyl-based CL" samples (light yellow) were compared with "transc-based CL and methyl-based MES" samples (dark yellow) for gene expression level (*EGFR*, *NF1*) and copy number (CN) segmentation level (*EGFR*, *CDKN2A*). All "transc-based CL and methyl-based MES" samples (dark yellow) show lower *EGFR* expression ($P = 0.08$, two-sided Wilcoxon rank sum test, the same for all following P values), lower *EGFR* amplification ($P = 0.0056$), higher *CDKN2A* amplification ($P = 0.015$), and lower *NF1* expression level ($P = 0.029$) than "transc-based CL and methyl-based CL" samples (light yellow). Data is available for $n = 20$ values of methyl-based CL-subtype, for $n = 5$ values of methyl-based MES-subtype in EGFR and NF1 expression panels, and for $n = 4$ values of methyl-based MES-subtype in CN seg value panels. Box plot center line represents median value, lower and upper hinges represent 25th and 75th percentiles, and lower and upper whiskers represent 1.5 interquartile ranges above and below box limits or maximum/minimum, whichever is closest to median. **C** "Transc-based MES and methyl-based CL" samples (light green) were compared with "transc-based MES and methyl-based MES" samples (dark green) for gene expression level (*EGFR*) and CN segmentation level (*EGFR*, *CDKN2A*). "Transc-based MES and methyl-based CL" samples (light green) show higher *EGFR* expression ($P = 0.04$), higher *EGFR* amplification ($P = 1.9 \times 10^{-4}$), and lower *CDKN2A* amplification ($P = 0.02$) compared to "transc-MES and methyl-based MES" samples (dark green). Data is available for $n = 9$ values of methyl-based CL-subtype in *EGFR* and *NF1* expression panels, for $n = 8$ values of methyl-based CL-subtype in CN seg value panels, for $n = 10$ values of methyl-based MES-subtype in *EGFR* and *NF1* expression panels, and for $n = 12$ values of methyl-based MES-subtype in CN seg value panels. Box plots are drawn as in (**B**). **D** Binary genetic alteration prediction results in the external validation set (NOA-04). **E** Confusion matrix of TCGA LGG samples based on gene expression subtype of transc-based prediction and methyl-based prediction. **F** Heatmap of TCGA-LGG samples ($n = 486$) with gene expression subtypes, histology, chr1p19q codel, *MGMT* promoter methylation, somatic mutations, and CNV. Statistical comparisons between genomic alterations and methyl- and transc-based gene expression is presented in Supplementary Data 3.

with single nucleotide polymorphisms (SNPs) within 10 bases of the targeted CpG site (snp-hit)[28]; (3) remove probes whose sequences align non-specifically (i.e. aligned to more than one location in the genome) (multi-hit)[28]; (4) remove probes not available on the current Infinium MethylationEPIC BeadChip array (EPIC, Illumina) in order to accommodate the application of this assay on the EPIC array. Then in a sample-wise fashion, probes with beadcount less than or equal to 3 and probes not significantly detected compared to the background (with a detection p value > 0.05) were set as missing values. Probes with more than 10% missing values across the samples were deleted. Samples with more than 5% missing values across all probes were deleted due to bad quality. The remaining missing values were imputed using the k-nearest neighbor (KNN) algorithm.

Ultimately, 1513, 2325, 2112, and 1279 probes were selected in the variable selection step for prediction of IDH mutation, *TERT*p mutation, *ATRX* mutation and 1p/19q co-deletion prediction, respectively.

For gene expression subtype, TCGA GBM samples with DNA methylation data from the Infinium HumanMethylation 27 K BeadChip array (HM27K, Illumina) or HM450K were included. GBM samples without gene expression data (Agilent 244 K) were excluded. TCGA level 3 data were directly used for samples with HM27K data. Probes used in the HM450K array included both Infinium I and Infinium II types, while probes used in the HM27K assay included only Infinium I type. The β-value derived from Infinium II probes has a smaller dynamic range and lower sensitivity compared to Infinium I probes[29]. Approximately 21,000 probes overlapped between the HM450K and HM27K platforms based on probe ID. However, the majority of overlapping probes from HM450K were Infinium II type probes while those from HM27K were mostly Infinium I probes, resulting in a likely batch effect when combining datasets. Batch adjustment was required to ensure the data were comparable between platforms. Several adjustment methods have been published, such as: BMIQ[30] (R package: ChAMP), SWAN[31] (R package: lumi), and PBC[32] (R package: wateRmelon). Acute myeloid leukemia (AML) DNA methylation data from TCGA was used for evaluating adjustment methods. All 194 AML samples were available on both HM27K and HM450K platforms. There are three steps to evaluate these methods, briefly described as below (Supplementary Fig. 4).

Step 1: 194 AML samples were clustered independently using the data from HM27K and HM450K. For each assay, probes were sorted by median absolute deviation (MAD) and the top 1000 and top 2000 probes were clustered with consensus non-negative matrix factorization[33] (CNMF) method. These two cluster results were compared and the concordance between these two platforms was high. This initial classification was used as the "gold standard."

Step 2: Three datasets that consisted of admixtures of data from the HM27K and HM450K datasets were simulated. Dataset 1 consisted of 25% samples coming from the HM27K dataset and 75% samples from the HM450K dataset. Dataset 2 consisted of 50% of samples from HM27K and 50% from HM450K. Dataset 3 consisted of 75% samples from HM27K and 25% from HM450K. CNMF clustering methods were applied on each of the admixture datasets. The clustering results from each admixture datasets were compared with the "gold standard."

Step 3: For each adjustment method, the admixture process was repeated and a fourth dataset containing only HM450K data was created. CNMF was applied to each admixture dataset and the membership for each sample was obtained. Membership indicated the subgroup which each sample belonged to, for example, the first AML sample may belong to subgroup 1 while the second AML sample belongs to subgroup 2. The membership of the classification was compared with gold standard as was done for the unadjusted datasets.

After the sample membership comparison, BMIQ showed the best results and was chosen as the final method for our data analysis. For more details and results, please refer to https://digitalcommons.library.tmc.edu/dissertations/AAI1597033/.

Just as for the binary genomic alteration prediction, for the gene expression subtype prediction, HM450K probes belonging to the following categories were deleted: missing from the EPIC platform, multi-hit[28], SNP-hit[28], located on chromosome X or Y, and with ≥5% missing values in the dataset. Then the retained 450K probes were intersected with 27K probes. Only probes existing in both platforms were kept for the following analysis. Retained probes belonging to the following categories were deleted: (1) with missing values in ≥5 samples; (2) probes not located on a CpG island; (3) probes not mapped to a known gene. For each probe, the Spearman correlation coefficient value was calculated between the methylation level and the corresponding gene's expression level among all samples. Probes with an absolute correlation coefficient value ≥ 0.1 were included (Fig. 5B).

After probe selection was completed, there remained 129 HM450K platform samples with 407067 probes and 287 HM27K platform samples with 23578 probes. After data integration, 416 samples with 20720 probes were available. After probe filters, 9519 probes were kept for correlation evaluation. We only kept the samples with gene expression information available, which led to 1263 probes and 356 samples as the final data set available. Of these, 212 samples in the training set were evaluated for their probe importance, 985 probes were assigned zero importance and then were excluded from the analysis, leaving 278 probes.

**Model Building: binary response variables.** For gene alterations within binary classes, TCGA annotations were used as the reference[34]: whole genome or exome sequencing (DNA-seq) for IDH and *ATRX* mutation; targeted sequencing or whole genome sequencing (DNA-seq) for *TERT*p mutation; and Affymetrix SNP6 array (SNP6) copy number variation (CNV) for chr1p19q codel. HM450K data were used as the independent input variables and represented as M-values[35]. Using IDH mutation as an example: samples were randomly apportioned into the training (60%), the development (20%), and the test (20%) sets, stratified by IDH mutation status. Variable selection and hyper-parameter tuning were applied within the training set using Elastic Net Regularization[36]. There are two parameters we need to specify: alpha ($\alpha$) and lambda ($\lambda$).

$$\frac{1-\alpha}{2}\left|\left|\beta\right|\right|_2^2 + \alpha||\beta||_1 \, (0 \leq \alpha \leq 1)$$

If alpha = 1, this is the lasso penalty; and if alpha = 0, it is the ridge penalty.

For the training set, the parameter alpha (R package: glmnet[36]) was set from 0.1 to 1, using 0.1 as a step. For each alpha value, 200 lambda values were randomly generated. Among the 200 lambda values, the best lambda value was picked out based on prediction accuracy. For each alpha and lambda value combination, 5-fold cross-validation (CV) was applied in the training dataset. For each fold among the cross validation, a set of probes was selected to build the regularized linear model with non-zero coefficient. By summarizing the selection results among 5 folds per CV, the percentage for each probe was calculated. Therefore, for each alpha value, we obtained a set of probes with their selection percentage. Then we combined the selected probe sets among ten alpha values and calculate their overall selection percentage. Probes were ranked by their selection percentage from high to low.

Based on the probe ranking, different top probe sets, which included different number of probes, were selected from high to low. For example, the top 100 probes, 200 probes, 500 probes, and so on were selected and form a probe set, respectively. For each probe set, a logistic regression model was refit with the training set. The alpha parameter was set from 0.1 to 1, with 0.1 as the step size, and for each alpha value, lambda was set from 0 to 5, with 0.05 as the step size. For each probe set, the best alpha and lambda value combination was picked out by prediction accuracy.

**A**

| | IDH | ATRX | TERTp | chr1p19q | #cases | CNS methylation-based calibrated predicted classes |
|---|---|---|---|---|---|---|
| Grp1 | Mutant | Mutant | WT | non-codel | 208 | A IDH(184) A IDH,HG(20) CONTR,HEMI(4) |
| Grp2 | Mutant | WT | Mutant | codel | 168 | O IDH(156) CONTR,HEMI(6) A IDH(3) A IDH,HG(3) |
| Grp3 | Mutant | WT | WT | non-codel | 41 | A IDH(33) A IDH,HG(6) CONTR,HEMI(2) |
| Grp4 | Mutant | WT | Mutant | non-codel | 7 | A IDH(4) CONTR,HEMI(3) |
| Grp5 | Mutant | WT | WT | codel | 3 | O IDH(3) |
| Grp6 | Mutant | Mutant | Mutant | non-codel | 1 | A IDH(1) |
| Grp7 | WT | WT | Mutant | non-codel | 176 | GBM,MES(71) GBM,RTK II (67) GBM,RTK I (24) CONTR,HEMI(9) CONTR,INFLAM(1) CONTR,REACT(1) GBM,RTK III (1) SFT,HMPC (1) SUBEPN,PF(1) |
| Grp8 | WT | WT | WT | non-codel | 39 | CONTR,HEMI(7) CONTR,CEBM(2) CONTR,WM(1) CONTR,INFLAM(2) CONTR,HYPTHAL(1) CONTR,PONS(1) GBM,G34(2) GBM,MID(2) GBM,MYCN(1) GBM<RTK II (1) LGG,GG(4) LGG,MYC(1) LGG,PA PF(2) LGG,PA/GG ST(1) A IDH,HG(1) ANA,PA(1) CNS,NB,FOXR2(3) IHG(2) DMG,K27(1) HGNET,BCOR(1) PLEX,PEDB(1) PTPR,B(1) |
| Grp9 | WT | Mutant | WT | non-codel | 1 | ANA PA(1) |

**B**

| Discordant samples | Detail information |
|---|---|
| All CONTR samples (40) | gliomas classifed as normal brain tissue, tumor purity can't explain (**C**) |
| Grp1:CONTR,HEMI(4) Grp2:CONTR,HEMI(6) Grp3:CONTR,HEMI(2) Grp4:CONTR,HEMI(3) | Classified as normal samples, expected to be IDH wildtype, but identified as IDH mutant samples (**A**) |
| Grp2:CONTR,HEMI(6) A IDH(3) A IDH,HG(3) | Do not expect chr1p19q co-del, but CNV profile shows codel (**D**) |
| Grp7: SFT,HMPC (1) | Expect euploidy genome, but shown chr10 del and chr19.20 amp (**E**) |
| Grp8: A IDH, HG (1) | Classified as IDH mutant category, but identified as IDH wild type (**A**) |
| Grp9: IHG (2) | IHG happens in infant. Two cases were diagnosed at age 45 and 53 yrs |

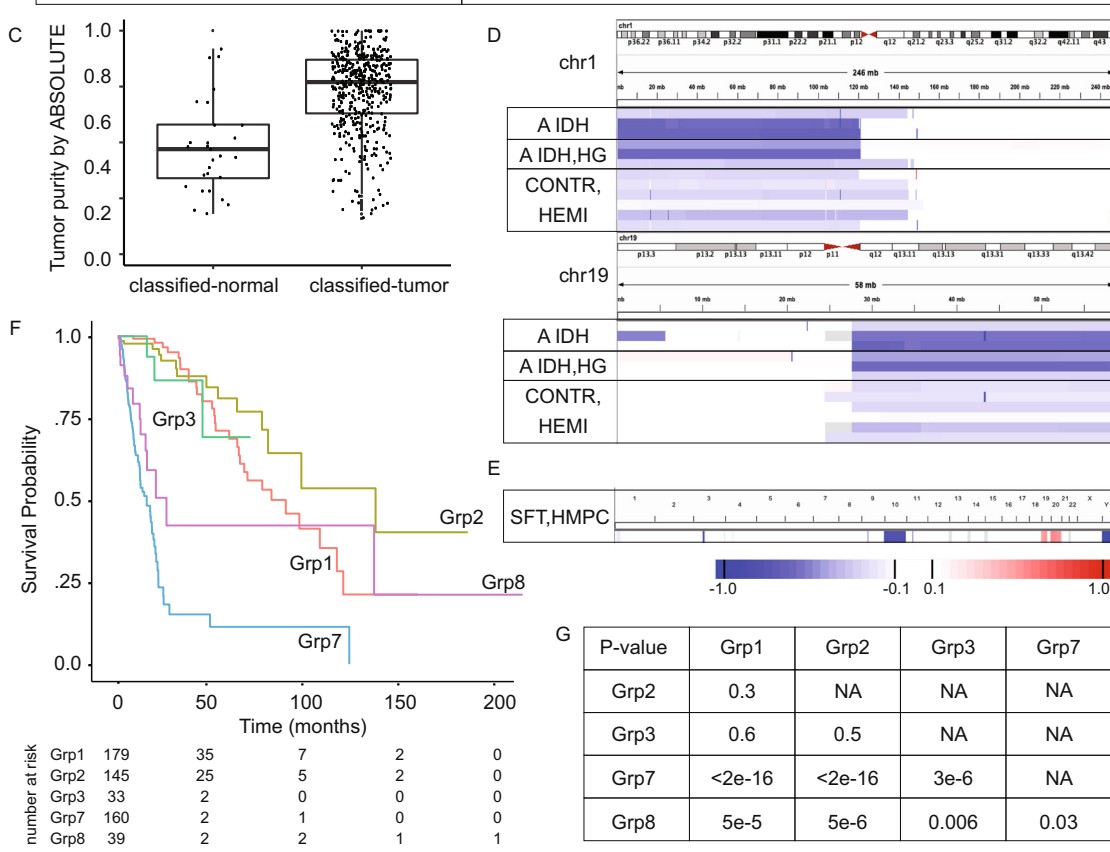

**G**

| P-value | Grp1 | Grp2 | Grp3 | Grp7 |
|---|---|---|---|---|
| Grp2 | 0.3 | NA | NA | NA |
| Grp3 | 0.6 | 0.5 | NA | NA |
| Grp7 | <2e-16 | <2e-16 | 3e-6 | NA |
| Grp8 | 5e-5 | 5e-6 | 0.006 | 0.03 |

The final model was determined based on performance in the development set; the number of probes used in the final models were 100, 1000, 500 and 100 for IDH mutation prediction, TERTp mutation prediction, ATRX mutation prediction and 1p/19q co-deletion prediction, respectively. The final alpha and lambda values were: for IDH mutation, TERTp mutation and ATRX mutation prediction, alpha = 0, lambda = 1; for 1p/19q co-deletion prediction, alpha = 0, lambda = 0.1. Performance of the models with varying parameter values are shown in Supplementary Figs. 5–8.

For validation, the final model was applied to the test set and the external validation set (NOA-04)[17]. This trial compared the efficacy and safety of radiotherapy followed by chemotherapy at progression to chemotherapy followed by radiotherapy at progression in patients with anaplastic gliomas (n = 115). DNA methylation HM450K data were available for all tumor samples. Most of the tumors were characterized for genomic alterations and these data served as the reference standard for comparison: targeted resequencing of the amplified

**Fig. 4 Classification comparison between UniD and Unsupervised CNS Classification. A** All samples with HM450K data available were classified into nine subgroups according to their methyl-based genetic alterations. The number of samples for each subset is also provided. The Unsupervised CNS Classification predicted categories are summarized for each subgroup in the rightmost column. **B** By comparing the predicted annotation between the UniD and Unsupervised CNS Classification, all discordant samples were picked out with detailed information. The left column shows the discordant samples and their subgroup belonging in (**A**). The right column shows the rationale for why they were discordant samples. **C** Boxplot shows the comparison of ABSOLUTE tumor purity between the samples classified into CONTR categories (classified-normal) and samples classified as tumor categories (classified-tumor) according to Unsupervised CNS Classification. Each point represents for one sample (classified-normal $n = 39$, classified-tumor $n = 565$). Boxplot center line represents median value, lower and upper hinges represent 25th and 75th percentiles, and lower and upper whiskers represent 1.5 interquartile ranges above and below box limits or maximum/minimum, whichever is closest to median. **D** SNP6-based CNV profile of the 12 samples in Grp2 (six CONTR, HEMI samples; three A IDH samples; and three A IDH, HG samples) clearly showing chr1p19q codel. The upper and lower panels show the profile of chr1 and chr19, respectively. **E** Whole genome CNV profile derived SNP6 array for the samples classified as SFT, HMPC in Grp7. **F** Kaplan–Meier plot with overall survival time (months) for the five enriched subgroups in (**A**) (Grp1, 2, 3, 7, and 8). The risk table is provided below. **G** Log-rank test was used to compare every two subgroups. $P$ values are provided in the table.

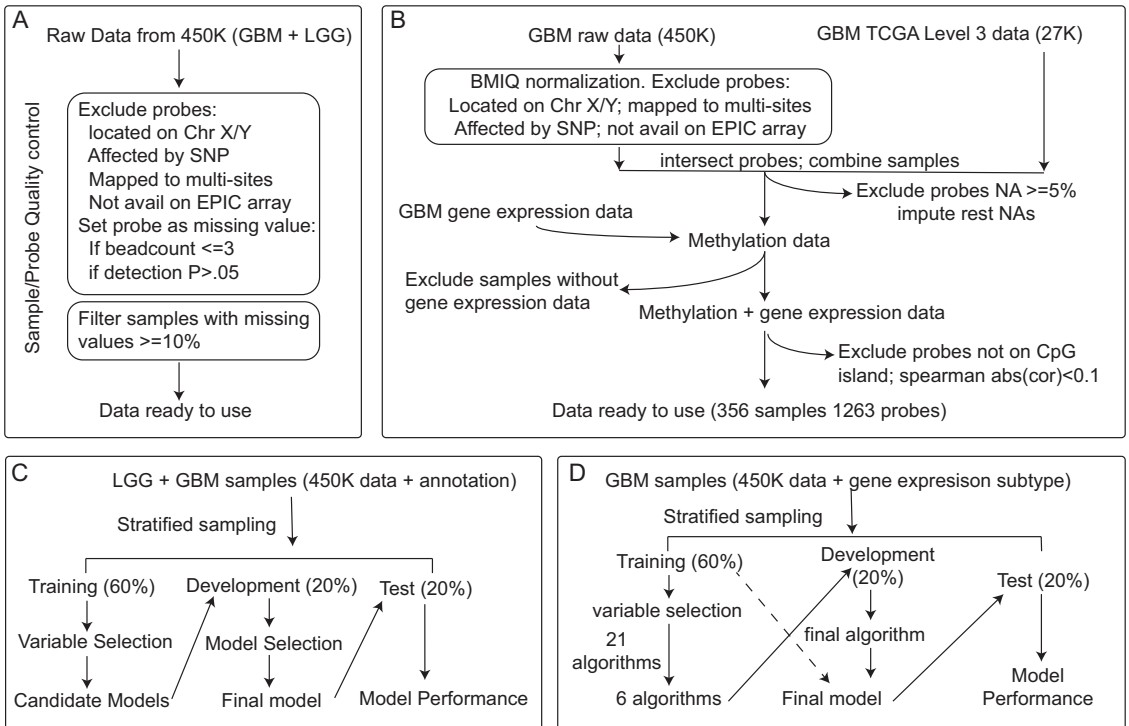

**Fig. 5 Workflow of data processing and model building. A** Data processing procedure for binary genetic alteration prediction with infiltrating gliomas. Step by step details are in Supplementary Methods section 1.1.1. **B** Data processing procedure for gene expression subtype prediction with GBM samples. Step by step details are in "Methods". **C** Model building procedure for binary genetic alteration prediction. Samples were randomly split into three sets: training set, development set, and test set. The training set was used for variable selection and to build candidate models, then the candidate models were applied to the development set. Based on the prediction accuracy of the development set, the final model was selected. The final model was applied to the test set for model performance evaluation. **D** Model building procedures for gene expression subtype prediction. GBM samples with DNA methylation and gene expression data available were included. DNA methylation probes were overlapped between the HM27K and HM450K platforms. Samples were split into training, development, and test set. Machine learning algorithms were evaluated on the training set and candidate algorithms were picked out. The development set was used to determine the final algorithm. The final model was built using the training and development sets and validated in the test set.

mutational hotspot (PCR-seq) for IDH ($n = 108$) and *TERT*p mutation ($n = 99$); multiplex ligation-dependent probe amplification (MLPA) for 1p19q codel ($n = 99$); and immunohistochemistry (IHC) for *ATRX* mutation ($n = 96$). In addition, IDH mutation status was also determined by unsupervised clustering with the HM450K data and chr1p19q codel status was also obtained by reviewing the CNV profiles derived from HM450K data ($n = 115$) (R package conumee)[20]. The methyl-based binary genetic alterations were predicted using build predicted models for all samples and compared to reference standards. The *MGMT* promoter methylation status obtained by methylation-specific PCR (MSP)[15] were compared with MGMT-STP27 prediction[13]. Note that the genomic alteration reference in the external validation set may be different from the training/development/test set. For example, chr1p19q codel status in the external validation set was annotated using either multiplex ligation-dependent probe amplification (MLPA) or direct CNV determination using HM450k data, while chr1p19q code status in the training/development/test set were determined using CNV derived from SNP6. Similar

model building strategies were applied to *TERT*p and *ATRX* mutation and chr1p19q codel (Fig. 5C) and a predictive model was built for each alteration. Accuracy was used as the major evaluation metric.

**Model building: gene expression subtype.** Samples were then randomly sampled into training (60%), development (20%), and test (20%) sets, stratified by gene expression subtype. For gene expression subtype prediction, the model building reference of subtype and each subtype's probability were calculated according to published algorithms[14]. Because reference samples might be heterogeneous and contain multiple subtypes, the probability of belonging to each subtype (Cl, PN, and MES) was calculated for every single sample using the formula shown below. For example, the probability that a given sample belongs to the CL subtype ($Prob_c$)

was calculated as:

$$\text{Prob}_C = \frac{1 - \text{P.value}_C}{(1 - \text{P.value}_C) + (1 - \text{P.value}_M) + (1 - \text{P.value}_P)}$$

$$= \frac{1 - \text{P.value}_C}{3 - (\text{P.value}_C + \text{P.value}_M + \text{P.value}_P)}$$

The permutation-based empirical $p$ value for each subtype was used. $P$ value$_c$ is the empirical $p$ value for the CL subtype, and similarly for other subtypes.

Probes were selected for model building based on entropy-based metrics, including information gain, gain ratio, and symmetrical uncertainty[22]. The R package Fselector (R package) built within the mlr package[37] was utilized to calculate those metrics. Probes were ranked by each evaluation metric from high to low. Then the rank sum was added up for each probe. Probes were sorted by the rank sum from high to low. The top probes were those which showed the most importance in terms of their response variable.

With the selected probes, 21 machine learning algorithms (Supplementary Table 6) were fitted and evaluated in the training set. With ranked probes, different probe sets were selected (top 100%, 90%, 80%, 70%, 60%, 50%, 40%, 30%, 20%, and 10% quantile) according to importance ranking (Supplementary Fig. 9). For each evaluated algorithm, a prediction model was fitted with a training set and a different probe set 100 times using fivefold CV (seed was set from 1 to 100). Models were evaluated using two metrics. First, the prediction accuracy was calculated by comparing the methylation data predicted gene expression subtype (predicted subtype) with the assigned subtypes obtained from gene expression data ("real" subtype) and determining the misclassification rate. Second, the sum of probability deviations was calculated by the sum of the square of probability deviations of each subtype, as shown below.

$$\text{prob.deviation}_C = (\text{real.probability}_C - \text{predicted.probability}_C)^2$$

$$\text{sum of probability deviations} = \text{prob.deviation}_C + \text{prob.deviation}_M + \text{prob.deviation}_P$$

For each algorithm and each selected probe set, the prediction accuracy and sum of probability deviations from 100 times of 5-fold CV were summarized. The misclassification rate of each machine learning algorithm and each probe set were summarized in Supplementary Fig. 10. The sum of probability deviations for each machine learning algorithm and each probe set was summarized in Supplementary Figs. 11 and 12. We can see that the prediction accuracy was not affected by the number of probes involved but the averaged sum of probability deviations decreased as the number of probes increased. Therefore, we use all 278 probes in order to maximize prediction accuracy.

Based on the evaluation, six top-performing candidate algorithms were applied to the development set (Supplementary Table 7). The final algorithm was selected based on the prediction accuracy in the development set (Fig. 5D). Random forest outperformed the others and was selected as the final algorithm. With the final algorithm determined, all samples from the training and development sets were used to build the final model. The final model was applied to the test set and TCGA LGG samples to evaluate its performance.

Tumor purity, derived by ABSOLUTE[38], was compared between the training, development, and test sets for each evaluated genomic alteration to avoid any potential bias. All sample sets showed no significant difference in tumor purity except for those used to develop the *ATRX* mutation predictor ($P = 0.043$, ANOVA test) (Supplementary Fig. 13).

**Statistical analysis**. For each binary genetic alteration, predictive signature probes were compared between the binary subgroups using the Wilcoxon rank sum test. The most significantly different probes were selected and applied to unsupervised clustering analysis. Predictive signatures of IDH, *TERT*p, and *ATRX* mutation, and 1p19q codel were further compared with the glioblastoma CpG island methylator phenotype (G-CIMP) signature[18].

Each signature probe set was mapped back to the reference genome and further compared for following categories. (1) *Chromosome enrichment*. The number of probes located on each chromosome was summarized and normalized by the total number of probes available on the chromosome. The percentage for each chromosome was calculated by the normalized percent. A proportional test (R function, prop.test) was applied between the number of probes for each chromosome and the total number of probes available for each chromosome. (2) *CpG island relationship enrichment*. Probes were categorized into following six classes based on their distance to CpG islands: CpG island, N_shlef (2–4 kb from island), S_shlef, N_shore (1–2 kb from island), S_shore, and unknown. The first class in order was used if probes were annotated with multiple categories. Then a proportional test was applied between the number of probes for each category and the total number of probes available for each category. (3) *Gene structure enrichment*. Probes were categorized into one of the following seven classes based on their relationship to functional gene structure: TSS200, TSS1500, Body, 3′-UTR, 5′-UTR, 1st Exon, and unknown. For probes that can be mapped to multiple gene structures, only the first category was used. The number of probes in each category was compared with the total available probes in each category. (4) *Mapping genes*. Genes mapped by the probes were summarized for frequency by array annotation.

If one probe mapped to multiple genes, then the first gene was counted. (5) *Gene Ontology (GO) enrichment*. Genes mapped by signature probes were used as input for GO analysis. This analysis was carried out with the web-based DAVID[19] (version 6.7).

Binary genetic alterations status was compared between methylation-based model prediction and the reference standard. To investigate the misclassified samples of *ATRX* mutation status, samples were regrouped by the DNA-seq and methylation-based (methyl-based) status. For misclassified samples, the single nucleotide variations (SNVs) called by MuTect2[39], VarScan[40], MuSE[41], and Somaticsniper[42] were collected (data obtained from Genomic Data Commons Data Portal[43]) and compared with the *ATRX* mutation status. To compare *ATRX* expression level between subgroups, the two-sample $t$ test was applied. To clarify the sample genetic characteristics, samples were regrouped by IDH, *ATRX*, and *TERT*p mutation status obtained by DNA-seq and methyl-based prediction. The methylation level (M-value) of HM450K probes located on *ATRX* were compared between subgroups using the ANOVA test. For 1p19q codel misclassified samples, sample CNV profiles were derived from HM450K data using the R package conumee[20].

For gene expression subtype prediction, misclassified samples in the test set ($n = 72$) were re-grouped by their transcriptional subtypes and methylation predicted subtypes. The correctly classified samples and misclassified samples were compared in terms of CNVs and gene expression for each transcriptional subtype using the Wilcoxon rank sum test.

For binary genetic alterations, the phase III clinical trial NOA-04[17] was used as an independent, external validation set. This trial compared the efficacy and safety of radiotherapy followed by chemotherapy at progression to chemotherapy followed by radiotherapy at progression in patients with anaplastic gliomas ($n = 115$). DNA methylation HM450K data were available for all tumor samples. Most of the tumors were characterized for genomic alterations and these data served as the reference standard for comparison: targeted resequencing of the amplified mutational hotspot (PCR-seq) for IDH ($n = 108$) and *TERT*p mutation ($n = 99$); multiplex ligation-dependent probe amplification (MLPA) for 1p19q codel ($n = 99$); and immunohistochemistry (IHC) for *ATRX* mutation ($n = 96$). In addition, IDH mutation status was also determined by unsupervised clustering with the HM450K data and chr1p19q codel status was also obtained by reviewing the CNV profiles derived from HM450K data ($n = 115$) (R package conumee)[20]. The methyl-based binary genetic alterations were predicted using build predicted models for all samples and compared to reference standards. The *MGMT* promoter methylation status obtained by methylation-specific PCR (MSP)[15] were compared with MGMT-STP27 prediction.

To validate the gene expression subtype prediction, TCGA LGG gene expression subtypes were compared between DNA methyl-based prediction and gene expression profile determined (transc-based)[14]. Histopathological and genomic characteristics were compared between methyl-based and transc-based subtype determinations using χ-square or Fisher's exact tests.

**Reporting summary**. Further information on research design is available in the Nature Research Reporting Summary linked to this article.

## Data availability

The data used for training the UniD algorithm include the TCGA glioblastoma data set, which is available in Genomic Data Commons Data Portal with project name as TCGA-GBM (https://portal.gdc.cancer.gov/projects/TCGA-GBM), and the TCGA low grade glioma data set, with project name TCGA-LGG (https://portal.gdc.cancer.gov/projects/TCGA-LGG). This data includes DNA methylation data, copy number variation data, transcriptome profiling data, and clinical information. The processed data are available within the Source Data file. The external validation data set from the NOA04 clinical trial that supports the findings of this study are available on request from the corresponding author of the paper "NOA-04 randomized phase III trial of sequential radiochemotherapy of anaplastic glioma with procarbazine, lomustine, and vincristine or temozolomide"[17]. This data includes DNA methylation data, copy number variation data, transcriptome profiling data, and clinical information. The remaining data are available within the Article, Supplementary Information or Source Data file. Source data are provided with this paper.

## Code availability

Data analysis and custom code were applied with R package (version 3.3). R packages utilized include conumee, ChAMP, lumi, wateRmelon, glmnet, Fselector, mlr, adabag, C50, party, earth, evtree, gbm, Rweka, kknn, kernlab, MASS, e1071, randomForest, randomForestSRC, ranger. All R packages are available from CRAN. To facilitate widespread adoption of the UniD platform, we developed an R package for rapid determination of biomarker status in gliomas (available on GitHub and the corresponding DOI is as follows https://doi.org/10.5281/zenodo.6563993)[44].

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

## Acknowledgements
Funding from R01 CA190121-01 (EPS) and the National Brain Tumor Society DefeatGBM Consortium (EPS, QW).

## Author contributions
J.Y., Q.W., and E.P.S. did the literature search and designed the study. J.Y., Y.M. did the data analysis. J.Y., B.W., and W.W. did the validation analysis. L.L. and R.E. did administration and management. J.Y., Q.W., Z.Z., J.M.K., A.T., M.S., J.H,. E.P.S. interpreted the data. J.Y. and E.P.S. wrote the manuscript. All authors have reviewed and revised the report.

## Competing interests
The authors declare no competing interests.
