## [Peer Review File · Nature Communications]

DNA-methylation based epigenetic signatures predict somatic genomic alterations in gliomasReviewers' Comments:

Reviewer #1:

Remarks to the Author:

The motivation of this study is not clear. The casual reader might not know what the motivation is of this study after reading the introduction.

The gene expression subtype classification needs a critical discussion as, several reports have shown that transcriptomic signatures in glioma are very heterogeneous and some studies have shown that some of the initial subtypes do not exist:

e.g. see: <https://www-pnas-org.stanford.idm.oclc.org/content/110/10/4009>

Therefore it is not clear how to judge this methylation based prediction of gene expression subtypes in the context of intra tumor transcriptomic heterogeneity.

It is not clear if the study is focused on glioblastoma (training of models) vs. application in both high and low grade glioma. The authors need to upfront make it clear if they are targeting all glioma or only glioblastoma.

The study is hard to follow and understand, several details are missing in terms of what the authors are doing, and why. To the average reader it will not be clear why certain analyses are done and how to interpret the results. E.g. what is the difference between UniD and Unsupervised CNS Classification?

How are the supervised classifiers for each of the genetic alterations turned into nine subset of glioma tumors?

The term deconvolutes might be poorly chosen. There is a rich literature on gene expression deconvolution, however, this manuscript has nothing to do with this. Therefore, it is important for the authors to clearly distinguish their work from this literature, or stop using the term "deconvolute" in this context.

Reviewer #2:

Remarks to the Author:

The manuscript presents an investigation in using epigenomic information (in the form of CpG methylation) to classify gliomas. The proposed pipeline is trained on TCGA data and evaluated on a new data set from a German clinical trial, showing very impressive results in particular in the prediction of binary genetic alterations, suggesting that DNA methylation is in fact an excellent marker in this case. Results on expression group prediction are somewhat less impressive but still indicated a good predictive power.

My focus on this review is on the methodological part; others will (hopefully) comment on the clinical significance of the proposed tool. From the statistical machine learning side, the project uses standard tools but it does so in a competent manner, partitioning the data sets appropriately into training, validation and test and offering some open source software which would be welcome. My main reservations are the somewhat limited discussion of which ones are the predictive features obtained via elastic net/ various regularisations applied. This is often an issue with predictors based on methylation (e.g. the famous clock) that the most relevant features do not lend themselves to ready interpretation. In this light, I think it would be useful for the authors to look into the robustness of the set of features obtained (I guess this is already done in part through the cross-validation process, so it would be good to see overlaps between features obtained in different CV runs). Related, would it be conceivably a good idea to develop targeted panels of few loci based on methylation? Similarly, I found the analysis of misclassifications less illuminating than anticipated by the authors ("Exploration... provides insights into the tumor biology...") in the intro. To me, it is a post-hoc

exercise which does not show empirically a robustness which would justify the claim of insights into biology.

Overall, the paper is well written with some minor grammatical lapses.

Reviewer #3:

Remarks to the Author:

DNA-methylation based epigenetic signatures predict and deconvolute somatic genomic alterations in gliomas

Overall, this is a well-designed, well-presented and complete study. It has the novelty of suggesting a new molecular classifier for glioma patients and it was extensively validated by using computational approaches, and an R package was built.

General comments:

I am not sure if this was intended to be a "methods" paper but biology was lacking: The work could benefit from more biological associations and conclusions in the main text; however, it is deeply immersed in the computational pipeline. For example, what would be the clinical implications of this in silico classification?

Instead of selecting one algorithm based on the best prediction accuracy, it might be better to adopt a combinatorial approach: Use the overlap of multiple algorithms. This removes the dependency on a single algorithm and the potential bias, as the best performing algorithm may change with the input training set, but the overlap of top performing ones should be robust to predict the TP.

Illumina 450K (in-house) and 27K BeadChip (TCGA) array data were used in the study. For some analyses (i.e. gene expression subtype prediction model building), it was restricted to the common probes between the two platforms. It would be interesting to see if this pipeline could be applied to platforms covering genome-wide Bisulfite-Seq datasets.

Do the investigated 4 genomic subtypes represent all glioma cases? How about other subtypes? Is this pipeline applicable to those? How would this classifier perform to predict a novel subtype not yet reported?

These 4 subtypes already show distinct genetic alterations and gene expression signatures, hence the methylation classifier reflects a high prediction accuracy in the test set. I would like to see the performance of this pipeline within more seemingly homogeneous group of samples in terms of mutations and expression, yet corresponding to different subtypes (i.e. presenting distinct clinical features), and would explore what other genomic and epigenomic features could be incorporated to better classify such cases.

The large sample size of the cohorts in this study allowed for division of the dataset into training, development and test validation sets, which is often a limitation in the application of machine learning algorithms.

REVIEWER COMMENTS

Reviewer #1, expert in methylation and prediction of gene expression (Remarks to the Author):

The motivation of this study is not clear. The casual reader might not know what the motivation is of this study after reading the introduction.

Author response: Thank you for this insightful comment. We have rephrased the introduction to make the primary purpose of this study clearer.

The key points for each paragraph that have been modified in the introduction are summarized below.

Paragraph 1: DNA methylation is important for us to understand the tumor biology. An unsupervised CNS classification was developed to classify CNS tumors into histopathological classes. This has become a widely adopted assay by neuropathology labs and illustrates the potential for methylation profiling.

Paragraph 2: Infiltrating gliomas have key molecular features which are critical for us to understand this devastating tumor better. Molecular features including:

Gene mutation of *IDH*, *ATRX*, *TERT* promoter

Copy number variations of chr1p19q

Gene expression subtypes: proneural, classical, and mesenchymal

Paragraph 3: Individual assays are required for each molecular feature mentioned above. The requirements of each individual assay can result in diagnosis delay and incomplete information. Therefore, we need a new platform which can provide all above molecular features in a rapid and cost-effective manner.

Paragraph 4: The UniD classifier was developed in our study to fulfill the need for a rapid and cost-effective test for gliomas which can elucidate the key genomic alterations in a clinical setting.

The gene expression subtype classification needs a critical discussion as, several reports have shown that transcriptomic signatures in glioma are very heterogeneous and some studies have shown that some of the initial subtypes do not exist:

e.g. see: <https://www-pnas-org.stanford.idm.oclc.org/content/110/10/4009>

Therefore it is not clear how to judge this methylation based prediction of gene expression subtypes in the context of intra tumor transcriptomic heterogeneity.

Author response: Thank you for this helpful suggestion. We agree that intratumor heterogeneity exists within glioblastoma not only at the gene expression, but also at the DNA mutation level. This is likely true in most solid tumors. Nevertheless, gene expression subtypes provide a powerful insight into the dominant tumor biology. It must be emphasized that the utility of gene expression subtype in guiding therapy has never been explored due to the lack of a clinical diagnostic for determining subtype. UniD addresses this limitation. The extent to which subtype-directed therapies will succeed or fail based on tumor heterogeneity remains an unanswered question and one beyond the scope of this manuscript.

The paper referenced in the review, entitled “Intratumor heterogeneity in human glioblastoma reflects cancer evolutionary dynamics” used the original TCGA **Verhaak classifier** to identify four gene expression subtypes (proneural, neural, classical, and mesenchymal) among the 51 tumor fragments from 10 patients and found that 6 of 10 cases have been classified into at least two different subtypes. While this may reflect tumor heterogeneity, it also reflects limitations of the original TCGA classification. Indeed, there is no clear definition at present for the four gene expression subtypes and signatures identified in the Verhaak classifier.

Our more recent study (Wang et al. 2017, Cancer cell, “Tumor evolution of glioma-intrinsic gene expression subtypes associates with immunological changes in the microenvironment”) has refined the subtypes and distinguished between tumor cell intrinsic and tumor microenvironment gene expression. In that analysis, we applied more rigorous transcriptomic analyses using glioma single cells, neurospheres, and tumor biopsies and identified three gene expression subtypes: proneural, classical, and mesenchymal. The neural subtype was not found using the single cell gene expression data, suggesting that the neural subtype signal was coming from the normal neuron cells. This study found that only a small subset of the TCGA samples (29/369, 8%) show multi-subtype activation. We have used our refined classification from Wang et al. to build the methylation classifier of gene expression.

It is not clear if the study is focused on glioblastoma (training of models) vs. application in both high and low grade glioma. The authors need to upfront make it clear if they are targeting all glioma or only glioblastoma.

Author response: Thank you for noting this potential area of confusion. Our study is designed for all infiltrating gliomas, including lower-grade glioma (diffuse gliomas) and high-grade gliomas (anaplastic and glioblastoma). We have summarized below what has been explained and clarified in the manuscript.

There are two categories of molecular features in terms of their status: binary genomic alterations and gene expression subtypes.

- (1) Binary genomic alterations (including IDH, *ATRX*, and *TERT*_p mutational status, and chromosome 1p19q co-deletion): Prediction models were developed using both high-grade and low-grade gliomas. Models can be applied to both high-grade and low-grade gliomas.
- (2) Gene expression subtypes (including Classical, Proneural, and Mesenchymal): Prediction models were developed using glioblastomas. This is because the gene expression subtypes were originally described using high-grade glioma datasets. In the “post-IDH” era, the majority of inter-tumor gene expression subtype heterogeneity is seen among IDH wild-type glioma (predominantly high-grade gliomas). Therefore, we developed our methylation predictor of gene expression subtype using high-grade gliomas. Nevertheless, the models could be applied to both high-grade and low-grade gliomas as demonstrated in our validation.

To clarify these points, we have made following edits in the manuscript:

1. From line 65 to line 69, we added an explanation about the two major categories: “*The above molecular features can be separated into two categories in terms of their status: binary class, including IDH, TERT_p, and ATRX mutation or wild type, and chr1p19q codel or intact; and gene expression subtypes, including CL, PN, or MES. Separate classifiers were developed for prediction of each of the binary classes (IDH, TERT_p, etc.) and for prediction of gene expression*”

subtype, using a rigorous machine learning approach.” This helps readers to understand the basic structure of Methods and Results section. We have added an explicit note (line 69-72) to point out that “The binary genomic alteration classifiers were trained and validated on a large cohort of both low-grade and high-grade glioma samples, while the gene expression subtype classifier was trained on glioblastoma samples only, since these subtypes were originally described using high-grade glioma datasets.”

2. From line 74 to line 76, we note that the validation cohort includes both low-grade and high-grade gliomas, and we emphasize that “The developed models can be easily applied to all infiltrating gliomas, including both low-grade gliomas and GBM”.

The study is hard to follow and understand, several details are missing in terms of what the authors are doing, and why. To the average reader it will not be clear why certain analyses are done and how to interpret the results. E.g. what is the difference between UniD and Unsupervised CNS Classification?

Author response: Thank you for pointing it out that the difference between UniD and Unsupervised CNS classification is not clear. We agree that some detailed information is missing in the manuscript and may not be clear to the reader. We provided some clarifications below and edited the manuscript accordingly.

UniD is a set of predictive models which can predict the genomic alteration status based on DNA methylation microarray data. The genomic statuses include:

- (1) gene mutation status of IDH, ATRX, and TERTp: mutant or wild type
- (2) copy number alteration status of chromosome 1p19q: co-deleted or intact
- (3) gene expression subtype: proneural, classical, or mesenchymal

Unsupervised CNS classification is the published classifier: DNA methylation-based classification of central nervous system tumors. This classifier focuses on distinguish central nervous system (CNS) tumors based on their DNA methylation profile.

The key differences between these two classifiers are:

1. Methodology difference.

UniD is based on a **supervised learning method**. We used the genomic alterations obtained from other conventional tests as the ground truth and build the predictive models.

Unsupervised CNS classification is built on an **unsupervised learning method**. It uses the unsupervised clustering approach to identify CNS tumor classes with distinct DNA methylation profiles.

2. Target objects difference.

UniD focuses only on gliomas.

Unsupervised CNS classification targets all CNS tumors.

3. Purpose difference.

The purpose of UniD is to provide specific genomic information. The output will be genomic alteration status of the given glioma sample, for example, whether the sample has IDH mutation or not.

The purpose of unsupervised CNS classification is to assist with CNS diagnosis in terms of histopathology. The output will be whether the CNS tumor belongs to embryonal tumor with multilayer rosettes, medulloblastoma, or diffuse midline glioma.

We have made the following edits in the manuscript to address this question:

1. Line 24 to line 26 in the revised manuscript: we added “*This unsupervised CNS classification used the unsupervised learning approach to identify CNS tumor classes with distinct DNA methylation profiles. The established random forest-based classifier can classify CNS tumor into one of those histopathological classes based on the tumor DNA methylation profile.*”
2. Line 230 to 234 in the revised manuscript: we added “*The key difference between the UniD and unsupervised CNS classification is that UniD aims to predict each infiltrating glioma’s key molecular features based on the DNA methylation values of selected loci for each molecular feature, while the Unsupervised CNS classification aims to classify each CNS tumor into one of the histopathological classes based on its overall DNA methylation profile. Accordingly, the Unsupervised CNS classification is an unsupervised learning-based model that has been developed to include all CNS tumors, while UniD is a supervised learning-based model which focuses only on gliomas. The objectives of these two classifiers are different, but it is informative to compare the results of these two classifiers.*”

How are the supervised classifiers for each of the genetic alterations turned into nine subset of glioma tumors?

Author response: Thank you for pointing this out. We have edited the manuscript in line 238 “*Gliomas(n=644) were classified into nine groups based on the UniD predicted molecular features status. These groups and their Unsupervised CNS classification-based classes were also summarized*” to make this point clearer. The nine subsets of glioma tumors are classified by the UniD predicted gene IDH, ATRX, TERTp gene mutation and chromosome 1p19q co-deletion status.

In Figure 5A table, from left to right, the second to fifth columns listed out the molecular feature status predicted by UniD. Based on their status, nine subsets of gliomas are defined. The main purpose for doing so is to compare the results between UniD classifier and the unsupervised CNS classification. The table in Figure 5B provides the details about discordant samples that we found when comparing the UniD predicted status and the Unsupervised CNS classification predicted classes.

The term deconvolutes might be poorly chosen. There is a rich literature on gene expression deconvolution, however, this manuscript has nothing to do with this. Therefore, it is important for the authors to clearly distinguish their work from this literature, or stop using the term ???deconvolute??? in this context.

Author response: Thank you for the good suggestion. We agree that the word “deconvolute” may not be the best choice to describe the focus of our study. Therefore, we have made the following changes in the manuscript:

1. Line 274 and 275: we changed the sentences from “*UniD accurately predicts ~~and deconvolutes~~ somatic genomic alterations in gliomas*” to “*UniD accurately predicts somatic genomic alterations in infiltrating gliomas*”.

2. Line 310: we changed the sentences from “*DNA methylation signatures accurately predict ~~and deconvolute~~ somatic genomic alterations*” to “*DNA methylation signatures accurately predict somatic genomic alterations*”.

We have also removed the word “deconvolute” from the title of the manuscript for the same reason.

Reviewer #2, expert in machine learning and epigenetics(Remarks to the Author):

The manuscript presents an investigation in using epigenomic information (in the form of CpG methylation) to classify gliomas. The proposed pipeline is trained on TCGA data and evaluated on a new data set from a German clinical trial, showing very impressive results in particular in the prediction of binary genetic alterations, suggesting that DNA methylation is in fact an excellent marker in this case. Results on expression group prediction are somewhat less impressive but still indicated a good predictive power.

Author response: Thank you for your positive and encouraging feedback. We agree that the prediction power of gene expression subtypes is less impressive, and we think this is because of the intrinsic heterogeneity of GBM, as noted above. Gene expression data obtained from tumor reflect a mixture of tumor cells and this issue is hard to avoid unless we use single-cell transcriptome data.

My focus on this review is on the methodological part; others will (hopefully) comment on the clinical significance of the proposed tool. From the statistical machine learning side, the project uses standard tools but it does so in a competent manner, partitioning the data sets appropriately into training, validation and test and offering some open source software which would be welcome.

Author response: Thank you for your positive feedback about the machine learning aspects.

My main reservations are the somewhat limited discussion of which ones are the predictive features obtained via elastic net/ various regularisations applied. This is often an issue with predictors based on methylation (e.g. the famous clock) that the most relevant features do not lend themselves to ready interpretation.

Author response: We agree that the interpretability of elastic net/regularization could be a potential limitation about the machine learning method we applied in this study.

In this light, I think it would be useful for the authors to look into the robustness of the set of features obtained (I guess this is already done in part through the cross-validation process, so it would be good to see overlaps between features obtained in different CV runs).

Author response: Thank you for bringing attention to this robustness issue. During our research, we have paid extra attention into this question. For those binary genomic alterations, there are 380011 probes available for selection. Among those probes, we applied different penalty levels (alpha in the glmnet package, equals to 0.1, 0.2, 0.3, up to 1) to the same training set and summarized the selected percentage for each probe. The selection percentage was averaged among all penalty levels results, and probes were sorted by this averaged selection percentage. Not surprisingly, most of the probes have never been selected. For example, for IDH mutation, 99.6% (378497/380011) probes have average selection percentages to zero. **Table 1** summarizes the average selection percentage information of binary genomic alterations.

For example, there are 100 probes involved in the IDH mutation predictive model. For these 100 probes, the mean, minimum, and maximum average selection percentages are 74.08%, 57.13%, and 100%, respectively. Among the 380011 probes, 1514 probes have been selected at least once in all iterated cross-validation processes. For those 1514 probes, the mean, minimum, and maximum average selection percentages are 13.5%, 0.01%, and 100%, respectively.

Table 1: average selection percentage information of binary genomic alterations

Genomic alterations	# of probes used in the predictive model	Mean (min, max) of average selection percentage for probes used in the predictive model	NO. of probes with average selection percentage>0	Mean (min, max) of average selection percentage for probes with positive average selection percentage
IDH	100	74.08% (57.13%, 100%)	1514	13.5% (0.01%, 100%)
ATRX	500	25.46% (11.79%, 99.04%)	2112	8.51% (2.7%, 99.04%)
TERTp	1000	13.71% (2.05%, 96.61%)	2326	6.35% (0.01%, 96.61%)
Chr 1p19q	100	63.59% (38.78%, 100%)	1280	9.97% (0.014%, 100%)

Moreover, for the gene expression subtype prediction, we selected 278 of the 1263 probes. For these 1263 probes, we calculated their importance using information.gain, symmetrical.uncertainty, and gain.ratio in the R package named mlr. Every single metric we used picked out 278 probes with positive importance while all remaining probes have zero importance. These consistent results increased our confidence about the selected 278 probes and the potential bias have been largely reduced.

Related, would it be conceivably a good idea to develop targeted panels of few loci based on methylation?

Author response: Thank you for this suggestion. We have thought about the possibility of developing a customized panel to target specific loci of DNA methylation. However, we found it difficult to persuade the production company to develop such a product. Moreover, commercial products (such as EPIC) not only cover what we need with relatively low cost, but more importantly also provide us with potential value in future studies. For example, as we collect more samples and test their whole-genome DNA methylation profiles, loci that were not used in this study could potentially be informative for future studies.

Similarly, I found the analysis of misclassifications less illuminating than anticipated by the authors ("Exploration... provides insights into the tumor biology...") in the intro. To me, it is a post-hoc exercise which does not show empirically a robustness which would justify the claim of insights into biology.

Author response: Thank you for bringing out this concern. We partially agree with your opinion that the conclusion made based on misclassification analysis was not very persuasive. However, we believe this misclassification analysis is necessary to make the study comprehensive and readers will be interested in this type of analysis.

To avoid any far-fetched conclusions based on the misclassification analysis, we have rephrased the following sentences:

1. Line 76 to 77. We changed the “*Exploration of DNA methylation-based misclassified cases provided insights into the biology of the tumor subtypes and demonstrated ...*” to “*Exploration of*

DNA methylation-based misclassified cases provided valuable ideas for future research directions and demonstrated...”

Overall, the paper is well written with some minor grammatical lapses.

Author response: Thank you for your positive feedback. We have proofread again to remove any grammar issues.

Reviewer #3, expert in glioma subtyping (Remarks to the Author):

DNA-methylation based epigenetic signatures predict and deconvolute somatic genomic alterations in gliomas

Overall, this is a well-designed, well-presented, and complete study. It has the novelty of suggesting a new molecular classifier for glioma patients and it was extensively validated by using computational approaches, and an R package was built.

Author response: Thank you for your positive feedback.

General comments:

I am not sure if this was intended to be a ???methods??? paper but biology was lacking: The work could benefit from more biological associations and conclusions in the main text; however, it is deeply immersed in the computational pipeline. For example, what would be the clinical implications of this in silico classification?

Author response: Thank you for pointing this out. We agree that this study is computationally intensive, and we tried hard to make sure the predictive models have good performance and applicability. In the results section, we provided some biological analysis, for example, in the sections entitled “prediction results analyses” and “model validation.” However, those analyses are still focused on the model results.

To answer your question, the clinical implication of this in silico classification is that this classifier can provide the status of multiple genomic alterations based on the DNA methylation profile. Currently, multiple assays are required to obtain the mutation status of IDH, *ATRX*, and *TERT*_p, copy number variation status of chromosome 1p19q, and gene expression subtypes. Now, with this UniD classifier, it is possible to obtain all the aforementioned genomic alterations by running only one assay, a DNA methylation assay. This point has been further emphasized in the fourth paragraph of Introduction (line 63 to line 79 in revised manuscript).

Instead of selecting one algorithm based on the best prediction accuracy, it might be better to adopt a combinatorial approach: Use the overlap of multiple algorithms. This removes the dependency on a single algorithm and the potential bias, as the best performing algorithm may change with the input training set, but the overlap of top performing ones should be robust to predict the TP.

Author response: Thank you for your great suggestion. We agree with you that a single algorithm may lead to biased selection results, which is why we applied the elastic net algorithm for feature selection of binary genomic alterations (including mutation status of IDH, *ATRX*, and *TERT*, and chromosome 1p19q code status). The elastic net algorithm is a combination of both lasso and ridge algorithms. This combination helped to balance the disadvantage of selection randomness (lasso) and the advantage of multicollinearity regularization (ridge). During the feature selection, we have applied different parameter value combinations to control the lasso and ridge which add different L1 and L2 penalty levels. Interestingly, different L1 and L2 penalty combinations lead to quite consistent results (**Table 1** in the manuscript). The table above summarizes the consistency of selected features that most of the probes we selected in our final predictive models have been selected in a high percentage of all cross-validation tests. Another important reason why we use elastic net is the fast speed of the algorithm since we have over hundreds of thousands of probes under consideration.

For the gene expression subtype feature selection (which selected 278/1263 probes), we combined the selection results from three algorithms: information gain, gain ratio, and symmetrical uncertainty. The selection results from these three algorithms are quite consistent: each algorithm picked out the same subset of probes (278 probes) with positive importance while all rest probes have importance equal to zero. This consistency helped us avoid the potential bias and increased our confidence in the final probes set.

Illumina 450K (in-house) and 27K BeadChip (TCGA) array data were used in the study. For some analyses (i.e. gene expression subtype prediction model building), it was restricted to the common probes between the two platforms. It would be interesting to see if this pipeline could be applied to platforms covering genome-wide Bisulfite-Seq datasets.

Author response: Thank you for this great suggestion. Theoretically, the pipeline can be applied to the selected DNA methylation loci derived from Bisulfite-Seq because the pipeline makes predictions based on the methylation level of selected loci. As long as the Bisulfite-Seq datasets covers the required DNA methylation loci, our framework should work.

Do the investigated 4 genomic subtypes represent all glioma cases? How about other subtypes? Is this pipeline applicable to those? How would this classifier perform to predict a novel subtype not yet reported?

Author response: Thank you for bringing out this concern. Basically, the 3 genomic subtypes (proneural, classical, and mesenchymal) we incorporated in this analysis covers the majority of gliomas. In fact, these three subtypes are mainly identified in the high-grade gliomas, glioblastoma, while most low-grade gliomas belong to the proneural subtype (this is shown in Figure 4E and 4F). We can see, based on the methylation-based subtype, that 422/486 low-grade gliomas are assigned to the proneural subtype. To the best of our knowledge, currently there is no other widely-accepted subtype identified in gliomas. But this definitely paves an avenue for future investigation.

In our algorithm, three numeric values are provided for each glioma which represent the probability for each subtype. The final subtype assignment is determined based on the highest probability of subtype. Based on the probability value, if any glioma has low probability values among three subtypes, then either it is an even mixture of three subtypes (intra-tumor heterogeneity) or it belongs to an unidentified new subtype. It would be interesting to validate this in the future if a novel subtype is identified.

These 4 subtypes already show distinct genetic alterations and gene expression signatures, hence the methylation classifier reflects a high prediction accuracy in the test set. I would like to see the performance of this pipeline within more seemingly homogeneous group of samples in terms of mutations and expression, yet corresponding to different subtypes (i.e. presenting distinct clinical features), and would explore what other genomic and epigenomic features could be incorporated to better classify such cases.

Author response: Thank you for pointing this out. This is actually a very interesting topic to explore. We have touched on it a little bit when comparing the results between the UniD classifier and unsupervised CNS classification. Until now, IDH mutation status and chromosome 1p19q status are the most important and well-known prognostic molecular feature in gliomas and they are already part of the WHO diagnosis criteria: glioma patients with IDH mutant or chromosome 1p19q code1 usually have better overall survival (OS). In Figure 5F and 5G, we found that IDH mutant glioma patients do not have significantly different survival regardless of their *ATRX*, *TERT*_p, and chr1p19 status (because Grp 1, 2, and 3 show no significant difference in log-rank test). Glioma patients with IDH wild type have significantly lower survival time than glioma patients with IDH mutation (Grp 7, and 8 versus Grp 1, 2, and 3). This indicates that IDH mutation status may be the leading feature which dominates the overall survival. Within samples with IDH wild type, patients with *TERT*_p mutant show worse survival compared to patients with *TERT*_p wild type (Grp 7 show worse survival curve than Grp8). By comparing the samples between Grp7 and Grp8, we found that Grp7 samples have significantly higher TERT expression level (Wilcoxon rank sum test, p-value=4.44e-10), higher mutation count (Wilcoxon rank sum test, p-value=5.15e-9), higher diagnosis age (Wilcoxon rank sum test, p-value=1.79e-9), and higher immune score calculated by ESTIMATE (Wilcoxon rank sum test, p-value=2.30e-5) than Grp8 samples (**Figure 1**). The grade does not show any significant difference (Fisher exact test, p-value=0.27) between the Grp7 (62.58%, 103/162) and Grp8 (52%, 13/25). Interestingly, as a favorable prognostic factor, the presence of *MGMT* promoter methylation shows significantly higher percentage (Fisher exact test, p-value=0.0036) in Grp7 (42.6%, 75/176) than Grp8 (27.67%, 7/39). It is possible that the effect of *TERT*_p mutation is stronger than *MGMT* promoter methylation.

Figure 1: Boxplot comparison between Grp7 and Grp8 samples

The large sample size of the cohorts in this study allowed for division of the dataset into training, development and test validation sets, which is often a limitation in the application of machine learning algorithms.

Author response: Thank you for your comments and we are lucky to have a large sample size in our study which is uncommon to see. The large sample size helped us to adopt the rigorous machine learning approach by leaving some samples for validation specifically.

Reviewers' Comments:

Reviewer #1:

Remarks to the Author:

the authors have addressed all my previous comments.

Reviewer #2:

Remarks to the Author:

I thank the authors for their considerate responses to my earlier criticism, and for the additional analyses. I am now happy with this manuscript.

Reviewer #3:

Remarks to the Author:

The authors have addressed all my concerns as Reviewer 3.